# Complete classification of steerability under local filters and its relation with measurement incompatibility

Huan-Yu Ku [1,2,3], Chung-Yun Hsieh [4] ✉, Shin-Liang Chen [1,5,6] ✉, Yueh-Nan Chen [1] ✉ & Costantino Budroni [2,3]

Quantum steering is a central resource for one-sided device-independent quantum information. It is manipulated via one-way local operations and classical communication, such as local filtering on the trusted party. Here, we provide a necessary and sufficient condition for a steering assemblage to be transformable into another via local filtering. We characterize the equivalence classes with respect to filters in terms of the steering equivalent observables (SEO), first proposed to connect the problem of steerability and measurement incompatibility. We provide an efficient method to compute the extractable steerability that is maximal via local filters and show that it coincides with the incompatibility of the SEO. Moreover, we show that there always exists a bipartite state that provides an assemblage with steerability equal to the incompatibility of the measurements on the untrusted party. Finally, we investigate the optimal success probability and rates for transformation protocols (distillation and dilution) in the single-shot scenario together with examples.

Einstein-Podolsky-Rosen steering[1–3] is a quantum correlation intermediate between entanglement[4] and Bell nonlocality[5,6]. A steering experiment consists of a remote state preparation where one party (Alice) prepares a local state for a distant party (Bob) by performing local measurements on her half of a bipartite entangled state and postselecting the outcome, which is communicated to Bob. As an interpretation in terms of classically postselected shared states is impossible, Alice seems to remotely steer the state of Bob.

In addition to being of foundational interest[7–14], due to the fact that only Bob is characterized, steering is at the core of one-sided (1S) device-independent (DI) quantum information processing[15–18]. A resource theory of steering was developed[19] to make sense of the manipulation of such resources, i.e., steerable state assemblages, for 1S-DI quantum information processing. A central open question is which state assemblages can be transformed into one another via the free operations allowed by resource theory, namely, one-way (1W) local operations and classical communication (LOCC). To date, this problem has been solved only for pure-qubit assemblages, which, in particular, has shown that there exist infinitely many equivalence classes and no measure-independent maximally steerable assemblage[19]. To make a parallel, this is a central problem in entanglement theory, where, e.g., entanglement distillation protocols were devised[20–22], and more generally, one is interested in the equivalence classes of entangled states reachable using stochastic LOCC, or local filtering[22–27]. This classification is already nontrivial in the three-qubit case, where two different classes arise[28], and infinitely many classes arise in multipartite settings with sufficiently high local dimension[29,30].

[1]Department of Physics and Center for Quantum Frontiers of Research & Technology (QFort), National Cheng Kung University, Tainan 701, Taiwan. [2]Faculty of Physics, University of Vienna, Boltzmanngasse 5, 1090 Vienna, Austria. [3]Institute for Quantum Optics and Quantum Information (IQOQI), Austrian Academy of Sciences, Boltzmanngasse 3, 1090 Vienna, Austria. [4]ICFO – Institut de Ciéncies Fotòniques, The Barcelona Institute of Science and Technology, Castelldefels 08860, Spain. [5]Dahlem Center for Complex Quantum Systems, Freie Universität Berlin, 14195 Berlin, Germany. [6]Department of Physics, National Chung Hsing University, Taichung 40227, Taiwan. ✉e-mail: andrew791006@gmail.com; shin.liang.chen@email.nchu.edu.tw; yuehnan@mail.ncku.edu.tw

Recently, the steering distillation problem was theoretically and experimentally addressed by Nery et al.[31]. They showed how to transform via local filtering a pure-qubit assemblage arising from measurements of $X$ and $Z$ on a partially entangled pure two-qubit state into another pure-qubit assemblage, arising from the same measurements on a maximally entangled bipartite state.

Quite surprisingly, a key ingredient to solve the steering assemblage classification problem is given by the notion of measurement incompatibility. Intuitively, measurement incompatibility refers to the impossibility of measuring certain physical quantities simultaneously, such as the position and momentum of a quantum particle (e.g., see refs. [32,33]). This property is at the foundation of many quantum phenomena, such as uncertainty relations[34], quantum contextuality[35–37], Bell nonlocality[38,39], and steering[40–42]. In particular, it has been shown that a state assemblage is unsteerable if and only if a collection of measurements, called steering-equivalent-observable measurement assemblage (SEO), is jointly measurable[43,44].

In this work, we provide an even stronger quantitative connection: (1) the SEO defines the equivalence classes of state assemblages and their transformations via local filtering and (2) its incompatibility is the maximal steerability over a class. With the concept of the equivalence classes and Alice's given measurements, a proper bipartite state $\rho_{AB}$ can be constructed such that the steerability of the resulting assemblage is the same as the incompatibility of such measurements. Finally, we provide an efficient method to compute the filter, analyse the success probability, and estimate the rate of the state assemblage transformation in the single-shot scenario.

## Results

### Quantum steering, measurement incompatibility, and steering-equivalent observables

We start with a brief summary of quantum steering, measurement incompatibility, and their relation. Given a bipartite state $\rho_{AB}$ shared between Alice and Bob, in each round of the steering protocol, Alice performs a measurement, labeled by $x$, on her half of the state and obtains a measurement result labeled by $a$ (see Fig. 1). The classical information $(x, a)$ is sent to Bob, who assigns this label to his state in that round. In quantum theory, each of Alice's measurements is represented by a positive-operator valued measure (POVM) $\{A_{a|x}\}_a$, where $A_{a|x} \geq 0$ and $\sum_a A_{a|x} = \mathbb{I}$[45]. Bob's state in each round can be computed as $\sigma_{a|x}/\mathrm{tr}(\sigma_{a|x})$, where $\sigma_{a|x} := \mathrm{tr}_A[(A_{a|x} \otimes \mathbb{I})\rho_{AB}]$. The collection of $\boldsymbol{\sigma} = \{\sigma_{a|x}\}_{a,x}$ is called the state assemblage. Similarly, the collection of POVMs $\boldsymbol{A} = \{A_{a|x}\}_{a,x}$ is called the measurement assemblage.

A state assemblage $\boldsymbol{\sigma}$ admits a local-hidden-state (LHS) model when it can be written as $\sigma_{a|x} = \sum_\lambda p(\lambda)p(a|x,\lambda)\rho_\lambda$; that is, it is obtained by postprocessing $\{p(a|x,\lambda)\}$ on a fixed collection of states $\{\rho_\lambda\}$ according to the distributions $\{p(\lambda)\}$. We denote the set of state assemblages admitting an LHS model as $\mathbb{LHS}$. State assemblages in $\mathbb{LHS}$ are called unsteerable, and steerable otherwise. Steering can be quantified via the steering robustness[46] defined as $\mathrm{SR}(\sigma) := \min\{t \geq 0 \mid \exists \boldsymbol{\xi}$ assemblage and $\boldsymbol{\tau} \in \mathbb{LHS}$ s.t. $(\sigma_{a|x} + t\xi_{a|x})/(1+t) = \tau_{a|x} \; \forall a, x\}$, and efficiently computed via semidefinite programming (SDP)[47].

Similar notions arise in the context of quantum measurements. Given a measurement assemblage $\{A_{a|x}\}_{a,x}$, it is said to be jointly measurable (JM) when all measurement effects can be interpreted as classical postprocessing of a single POVM $\{G_\lambda\}_\lambda$, namely $A_{a|x} = \sum_\lambda p(a|x, \lambda)G_\lambda$. If that is not the case, it is said to be incompatible. A measure of incompatibility, the incompatibility robustness[43,48,49], can be defined as $\mathrm{IR}(\boldsymbol{A}) = \min\{t \geq 0 \mid \exists \boldsymbol{N}$ measurement assemblage and $\boldsymbol{D} \in \mathbb{JM}$ s.t. $(A_{a|x} + tN_{a|x})/(1+t) = D_{a|x} \; \forall a,x\}$, where $\mathbb{JM}$ denotes the set of jointly measurable measurement assemblages.

These similarities are not accidental: It has been shown that there exists a strong connection between steerability and incompatibility[40,41] and even that there is a one-to-one mapping between the two mathematical problems[43] (see ref. [44] for the infinite-dimensional case). The

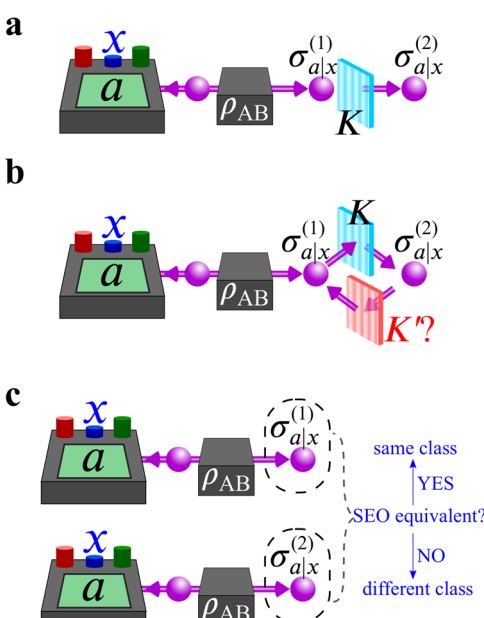

**Fig. 1 | Schematic illustration of this work.** In this work, we consider three fundamental quantum information scenarios: that is, distillation, convertibility, and classification, in a steering-type scenario, where Alice measures her part of the system on state $\rho_{AB}$ and remotely projects Bob's systems into a collection of (subnormalized) states $\sigma_{a|x}$. **a** In the distillation scenario, one asks how much steerability can be distilled by a local filter $K$, that maps $\sigma_{a|x}^{(1)}$ to $\sigma_{a|x}^{(2)}$. **b** In the convertibility scenario, one looks for the existence of a filter $K'$ mapping $\sigma_{a|x}^{(2)}$ back to $\sigma_{a|x}^{(1)}$. **c** In the classification scenario, one classifies different assemblages into the same class if they belong to the same steering-equivalent observable (SEO). By showing the equivalence between the convertibility and classification problems [scenarios (**b**) and (**c**)], we are able to obtain the optimal filter that distills the maximal steerability from $\sigma_{a|x}^{(1)}$ to $\sigma_{a|x}^{(2)}$ [scenario (**a**)].

mathematical equivalence is introduced via the notion of steering-equivalent-observable measurement assemblage (SEO)[43]: a state assemblage $\boldsymbol{\sigma}$ is steerable if and only if the measurement assemblage of its SEO $\boldsymbol{B}$ is incompatible. To define SEO $\boldsymbol{B}$, we need to restrict the reduced state $\rho_B := \sum_a \sigma_{a|x}$ to its range $\mathscr{K} := \mathrm{ran}(\rho_B)$ via the projection $\Pi_B : \mathscr{H}_B \to \mathscr{K}$, where $\Pi_B\Pi_B^* = \mathbb{I}_\mathscr{K}$ and $\Pi_B^*\Pi_B$ is a Hermitian projector in $\mathcal{L}(\mathscr{H}_B)$. Then, we define the reduced state and state assemblage restricted to $\mathscr{K}$ as, respectively, $\tilde{\rho}_B := \Pi_B\rho_B\Pi_B^*$ and $\tilde{\sigma}_{a|x} := \Pi_B\sigma_{a|x}\Pi_B^*$, respectively. In the following, we use the notation ~ to denote an assemblage restricted to the range of the corresponding reduced state.

Then, $\boldsymbol{B}$ is defined as

$$B_{a|x} := \tilde{\rho}_B^{-\frac{1}{2}}\tilde{\sigma}_{a|x}\tilde{\rho}_B^{-\frac{1}{2}}. \tag{1}$$

This allows the SEO to be well-defined even when $\rho_B$ is not full-rank[43]. With a slight abuse of notation, we write $\rho_B^{-\frac{1}{2}} := \tilde{\rho}_B^{-\frac{1}{2}} \oplus 0_{\mathscr{K}^\perp}$, to denote the embedding into the original space $\mathscr{H}_B = \mathscr{K} \oplus \mathscr{K}^\perp$, where $\perp$ is the orthogonal complement.

### Transforming state assemblages via local filters

First, we introduce an equivalence relation between two state assemblages, $\boldsymbol{\sigma}^{(1)}$ and $\boldsymbol{\sigma}^{(2)}$, based on their SEOs. We define the equivalence relation ~$_{SEO}$ as follows:

$$\boldsymbol{\sigma}^{(1)} \sim_{SEO} \boldsymbol{\sigma}^{(2)} \overset{\mathrm{def}}{\Longleftrightarrow} B_{a|x}^{(1)} \oplus 0_{\mathscr{K}_{(1)}^\perp} = U\left(B_{a|x}^{(2)} \oplus 0_{\mathscr{K}_{(2)}^\perp}\right)U^\dagger \; \forall a,x, \tag{2}$$

where $\mathscr{K}_{(i)} := \mathrm{ran}(\rho_B^{(i)})$ for $i = 1, 2$, and $U$ is a unitary operator acting on $\mathscr{H}_B$. This definition requires that $\mathscr{K}_{(1)}$ and $\mathscr{K}_{(2)}$ are isomorphic and that

the two SEOs $B^{(1)}$ and $B^{(2)}$ are the same up to a local change of basis. It is straightforward to see that $\sim_{\mathrm{SEO}}$ is an equivalence relation, namely, it is reflexive, symmetric, and transitive. Hence, it gives rise to equivalence classes, which we denote by $[\boldsymbol{\sigma}]$.

We now introduce another type of steering class based on transformation by local filters. Local filters on Bob's side are modeled via the map

$$\sigma_{a|x} \mapsto \frac{K\sigma_{a|x}K^{\dagger}}{p_{\mathrm{succ}}}, \quad \forall a,x, \tag{3}$$

where $K$ satisfies $K^{\dagger}K \leq \mathbb{I}$ and $p_{\mathrm{succ}} := \mathrm{tr}[\sum_a \sigma_{a|x}K^{\dagger}K]$. In the case $p_{\mathrm{succ}} = 0$, one could define the output of the map as the operator 0. Of course, the transformation makes sense only if $p_{\mathrm{succ}} > 0$, otherwise the transformation is simply impossible. This corresponds to making a local measurement and postselecting a specific outcome. In the language of the one-way (1W) stochastic (S) local operations and classical communication (LOCC), or 1W-SLOCC operations[19,50], these are the most general local filters, which are denoted as $\mathrm{LF}_1$ to emphasize that they contain only one Kraus operator. See Supplementary Note 1 for a self-contained summary. In addition, 1W-SLOCC also contains a classical pre and postprocessing on Alice's side, which is not considered here (also see Supplemental Note 1).

It is convenient to introduce some notation to denote the existence of such a transformation, we write

$$\boldsymbol{\sigma}^{(1)} \xrightarrow{\mathrm{LF}_1} \boldsymbol{\sigma}^{(2)} \text{ if } \boldsymbol{\sigma}^{(1)} \text{ transformable into } \boldsymbol{\sigma}^{(2)} \text{ via } \mathrm{LF}_1. \tag{4}$$

Similar to SEO, $\mathrm{LF}_1$ filters define an equivalence relation. We define $\sim_{\mathrm{LF}_1}$ as

$$\boldsymbol{\sigma}^{(1)} \sim_{\mathrm{LF}_1} \boldsymbol{\sigma}^{(2)} \xleftrightarrow{\mathrm{def}} \boldsymbol{\sigma}^{(1)} \xrightarrow{\mathrm{LF}_1} \boldsymbol{\sigma}^{(2)} \text{ and } \boldsymbol{\sigma}^{(2)} \xrightarrow{\mathrm{LF}_1} \boldsymbol{\sigma}^{(1)}. \tag{5}$$

Clearly, this is an equivalence relation, i.e., reflexive, symmetric, and transitive. Hence, it gives rise to another set of equivalence classes. We can now connect these two notions through the following theorem:

**Theorem 1.** Consider two assemblages $\boldsymbol{\sigma}^{(1)}$ and $\boldsymbol{\sigma}^{(2)}$. Denote their reduced states as $\rho^{(i)} := \sum_a \sigma_{a|x}^{(i)}$, their ranges as $\mathscr{K}_i := \mathrm{ran}(\rho^{(i)})$, and the dimensions as $d_i := \dim(\mathscr{K}_i)$, for $i = 1, 2$. Then, the following statements are equivalent
(i) $\boldsymbol{\sigma}^{(1)} \sim_{\mathrm{SEO}} \boldsymbol{\sigma}^{(2)}$
(ii) $\boldsymbol{\sigma}^{(1)} \sim_{\mathrm{LF}_1} \boldsymbol{\sigma}^{(2)}$
(iii) $\boldsymbol{\sigma}^{(2)} \xrightarrow{\mathrm{LF}_1} \boldsymbol{\sigma}^{(1)}$ and $d_1 = d_2$.

Moreover, in the case $\boldsymbol{\sigma}^{(1)} \sim_{\mathrm{SEO}} \boldsymbol{\sigma}^{(2)}$, the filter $K$ can be explicitly computed as a function of the reduced states $\rho^{(i)} = \sum_a \sigma_{a|x}^{(i)}$ and the unitary $U$ appearing in Eq. (2). Such a filter can be constructed to have the success probability

$$p_{\mathrm{succ}} = \left[ \lambda_{\max} \left( \rho^{(2)-1/2} U^{\dagger} \rho^{(1)} U \rho^{(2)-1/2} \right) \right]^{-1}, \tag{6}$$

where $U$ is the unitary appearing in Eq. (2) and $\lambda_{\max}(X)$ denotes the maximum eigenvalue of the operator $X$. This value is provably optimal if the initial assemblage contains sufficiently many linearly independent states to perform channel tomography.

A detailed proof is presented in the Methods section.

Theorem 1 connects two seemingly distinct concepts: equivalence classes with respect to SEO and with respect to $\mathrm{LF}_1$. Thus, they provide a new physical interpretation of the SEOs beyond the one-to-one mapping of steerability into incompatibility[43]: SEOs classify all assemblages with respect to $\mathrm{LF}_1$ local filters in the sense that whether the two assemblages can be converted to each other by $\mathrm{LF}_1$ is determined by

their SEOs. Moreover, Theorem 1 provides a simple necessary and sufficient condition for the existence of a reverse transformation. Namely, given the transformation from $\boldsymbol{\sigma}^{(2)}$ to $\boldsymbol{\sigma}^{(1)}$, the reverse transformation from $\boldsymbol{\sigma}^{(1)}$ to $\boldsymbol{\sigma}^{(2)}$ exists if and only if the ranks of $\rho^{(1)}$ and $\rho^{(2)}$ are the same. Thus, transformable assemblages of the same rank can always be discussed in terms of equivalence classes with respect to two-way transformations.

In this sense, we can define a canonical representative assemblage of each equivalence class $[\boldsymbol{\sigma}]$ as

$$\sigma_{a|x}^{\boldsymbol{B}} := B_{a|x}/d \tag{7}$$

with $\boldsymbol{B}$ the SEO of $\boldsymbol{\sigma}$ and $d = \dim(\mathrm{ran}\rho)$ the rank of the reduced state $\rho = \sum_a \sigma_{a|x}$. It is clear that all the assemblages in this class can be transformed into the canonical-state assemblage with the transformation in Eq. (1). As we will demonstrate below, this interpretation can be further expanded.

## Maximal and minimal robustness within each class

Here we present a general result on the minimal and maximal robustness that can be achieved via $\mathrm{LF}_1$ local filters.

**Theorem 2.** Given a state assemblage $\boldsymbol{\sigma}$, its corresponding SEO $\boldsymbol{B}$, and its equivalence class $[\boldsymbol{\sigma}]$ (w.r.t. $\sim_{\mathrm{LF}_1}$), we have

$$\mathrm{SR}^{\sup}([\boldsymbol{\sigma}]) := \sup_{\boldsymbol{\sigma}' \sim_{\mathrm{LF}_1} \boldsymbol{\sigma}} \mathrm{SR}(\boldsymbol{\sigma}') = \mathrm{IR}(\boldsymbol{B}), \tag{8}$$

$$\mathrm{SR}^{\inf}([\boldsymbol{\sigma}]) := \inf_{\boldsymbol{\sigma}'' \sim_{\mathrm{LF}_1} \boldsymbol{\sigma}} \mathrm{SR}(\boldsymbol{\sigma}'') = 0. \tag{9}$$

Moreover, for any $\varepsilon > 0$, one can efficiently find a filter (via SDP) that transforms $\boldsymbol{\sigma}$ into the assemblage $\boldsymbol{\sigma}'$ such that $\mathrm{SR}(\boldsymbol{\sigma}') \geq \mathrm{IR}(\boldsymbol{B}) - \varepsilon$, as in Eq. (8), and one that transforms it into the assemblage $\boldsymbol{\sigma}''$ such that $\mathrm{SR}(\boldsymbol{\sigma}'') \leq \varepsilon$, as in Eq. (9), by a direct calculation.

A detailed proof of Theorem 2 can be found in the Methods section, together with the description of the SDP. Intuitively, the result on the sup comes from equating the SDP definition of $\mathrm{IR}(\boldsymbol{B})$ with optimization over the SEO for $\mathrm{SR}(\boldsymbol{\sigma}')$, whereas the result on the inf comes from the fact that one can transform any assemblage into one coming from a pure state with arbitrary low entanglement. Notice the use of sup/inf instead of max/min. Even though this is a fundamental difference at the mathematical level, in the sense that the exact bound may be unreachable, every physical experiment will always have some nonzero uncertainty, making this difference irrelevant. The same argument applies to numerical computations, such as those of SDPs.

It is interesting to notice that the assemblage giving the maximal steerability in a given equivalence class is not necessarily the canonical representative $\boldsymbol{\sigma}^{\boldsymbol{B}}$ of Eq. (7), which is generated by sharing a maximally entangled state. A more detailed discussion is presented in Th. 3 and an explicit counterexample is provided in Supplementary Note 3.

The results of Theorem 2, combined with those of Theorem 1, further extend the new interpretation of SEOs. In fact, they not only characterize the equivalence classes w.r.t. $\mathrm{LF}_1$ filters but also provide a tight bound on the maximal steerability within each class. Moreover, one can saturate the previous inequality $\mathrm{SR}(\boldsymbol{\sigma}) \leq \mathrm{IR}(\boldsymbol{B})$, derived in ref. 51 (see also refs. 52,53), if local filters are allowed.

A second observation is that rather counterintuitively, the same equivalence class contains assemblages that have maximal and arbitrarily small steerability. In one direction this may be obvious, as one can always decrease steerability by means of local operation, e.g., by decreasing the amount of entanglement in the initial state. In the other direction, the physical soundness of this result is recovered by noticing that even if an assemblage can be transformed into a maximally steerable one, this happens with vanishing probability. This can be

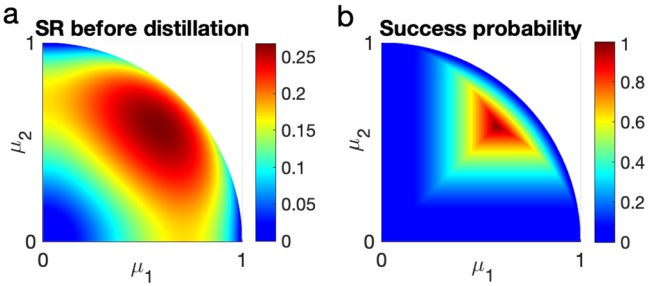

**Fig. 2 | Results of the qutrit assemblage before and after the filter with the success probability. a** Steering robustness SR of the qutrit assemblages before the filter. Here, the qutrit assemblages are generated by the purely entangled qutirt state $|\psi\rangle = \sum_i \mu_i |ii\rangle$ satisfying $\sum_i \mu_i^2 = 1$ and $1 > \mu_i > 0\ \forall\ i$ with Alice's measurements being two mutually unbiased bases in dimension three. **b** Success probability of distilling qutrit assemblages using Eq. (11). After the filter protocol, the steering robustness is 0.2679 $\forall\ \mu_1$, and $\mu_2$. The white region represents the nonphysical case because $\mu_1^2 + \mu_2^2 \geq 1$ cannot be satisfied.

seen, for instance, in the explicit construction used in the proof (see Methods section and the example in Fig. 2).

**Optimal state with a given measurement assemblage on the untrusted side**
We first state the main result:

**Theorem 3.** For any measurement assemblage $\boldsymbol{A}$ and any $\varepsilon > 0$, via SDP we can efficiently compute a bipartite state $\rho_{AB}^\varepsilon$, that generates an assemblage $\sigma_{a|x} := \mathrm{tr}_A[(A_{a|x} \otimes \mathbb{I})\rho_{AB}^\varepsilon]$ satisfying $\mathrm{SR}(\boldsymbol{\sigma}) \geq \mathrm{IR}(\boldsymbol{A}) - \varepsilon$.

Details of the proof and an explicit construction via SDP of the bipartite state are presented in Supplementary Note 2. Interestingly, the bipartite state providing maximum steerability is not necessarily maximally entangled. In detail, given the maximally entangled state and the measurement assemblage $\boldsymbol{A}$ used on Alice's side to generate Bob's state assemblage $\boldsymbol{\sigma}$, we have the SEO $\boldsymbol{B} = \boldsymbol{A}$ and the state assemblage $\boldsymbol{\sigma} = \boldsymbol{\sigma}^B$. We present an explicit example such that $\mathrm{IR}(\boldsymbol{B}) > \mathrm{SR}(\boldsymbol{\sigma}^B)$ in Supplementary Note 3. It is also interesting to recall the following inequality derived by ref. 51: $\mathrm{SR}(\boldsymbol{\sigma}) \leq \mathrm{IR}(\boldsymbol{B}) \leq \mathrm{IR}(\boldsymbol{A})$. Theorem 3, then, tells us that this bound is saturated, i.e., given a measurement assemblage, we can always find a bipartite state such that the steerability of the associated assemblage coincides with the incompatibility of the original measurements. Finally, this result is outside the 1S-DI framework, as it requires the knowledge of Alice's measurements and of the bipartite state. To highlight this difference, notice that given a state assemblage in the class associated with an SEO $\boldsymbol{B}$, such that $\mathrm{IR}(\boldsymbol{B}) < \mathrm{IR}(\boldsymbol{A})$, there is no way to increase its steering robustness up to $\mathrm{IR}(\boldsymbol{A})$ via $\mathrm{LF}_1$ filters due to Theorems 1 and 2.

**Conversion rates between assemblages**
Local filter corresponds to a local measurement performed on Bob's system. In the case of a successful outcome, the system is kept; otherwise, it is discarded. A key figure of merit is, thus, the rate at which the target assemblages are produced. More precisely, the rate $r$ at which one transforms an assemblage $\boldsymbol{\sigma}$ into another assemblage $\boldsymbol{\sigma}^*$ can be defined in terms of the existence of a transformation[31]

$$(\boldsymbol{\sigma})^{\otimes N} \xrightarrow{\text{1W-SLOCC}} (\boldsymbol{\sigma}^*)^{\otimes rN}, \tag{10}$$

with probability 1 in the limit of $N \to \infty$ and with $0 < r \leq 1$. In principle, this definition allows for the use of global operations on multiple copies of the assemblage, i.e., $(\boldsymbol{\sigma})^{\otimes N}$. However, our local filter method

can be formulated for a single-shot scenario. In other words, given a single copy of a state assemblage $\boldsymbol{\sigma}$, there is a nonzero probability of transforming it into the target assemblage $\boldsymbol{\sigma}^*$. In this case, the rate $r$ is the single-shot success probability:

$$r = p_{\text{succ}} = \mathrm{tr}[\rho_B K^\dagger K], \tag{11}$$

where $\rho_B = \sum_a \sigma_{a|x}$ is the reduced state on Bob's side, and $K$ is the filter. See Supplementary Note 4 for details.

**Application to qutrit assemblages**
The first observation is that the results of the pure-qubit case by ref. 31 are recovered through our formalism. For completeness, these results are rederived in our language in Supplementary Note 5. Here, we provide an example of a qutrit system with two inputs and three outputs. Consider the two-qutrit state $|\psi\rangle = \sum_{i=1}^3 \mu_i |ii\rangle$ with $\mu_i > 0\ \forall\ i$ and $\sum_{i=1}^3 \mu_i^2 = 1$. Denote the reduced state by $\tau = \sum_{i=1}^3 \mu_i^2 |i\rangle\langle i|$ and the minimal eigenvalue of $\tau$ by $\lambda_{\min}(\tau) = \min_i \mu_i^2$. We choose Alice's measurement assemblage to contain the measurements in the computational basis and its Fourier transform; namely, $\{A_{a|0}\} = \{|a\rangle\langle a|\}$ and $\{A_{a|1}\} = \{F|a\rangle\langle a|F^\dagger\}$ with $a = \{1, 2, 3\}$. Here, $F$ is the three-dimensional discrete Fourier transform. The corresponding measurement bases are mutually unbiased. The initial assemblage is $\sigma_{a|x} = \tau^{1/2} A_{a|x} \tau^{1/2}\ \forall\ a,x$. Via SDP one can compute the optimal assemblage in this class, to obtain $\boldsymbol{\sigma}^* = \boldsymbol{A}^T/3$. Consequently, the local filter is $K := \sqrt{3\lambda_{\min}(\tau)} \tau^{-1/2}$. A conversion rate of $r = p_{\text{succ}} = 3\lambda_{\min}(\tau)$ is then obtained. We note that this optimal assemblage provides not only the maximal steering robustness but also the maximal randomness generation[17] in the sense that $\mathrm{tr}(\sigma_{a|1}) = 1/3\ \forall\ a$. We visualize the values of SR and success probability in Fig. 2. Finally, we recall the discussion below Theorem 2. In this example, there exists an assemblage with vanishing steerability that can be transformed into the maximally steerable one in this class.

## Discussion
This work investigated the convertibility between state assemblages via local filters on the trusted party (Bob). These local filters, denoted as $\mathrm{LF}_1$, are sufficient to generate the most general 1W-SLOCC operations when combined with classical pre and postprocessing on Alice's side[19]. Note that local filters do not introduce any loophole in the steering scenario. This is because a local filter can be performed as a part of the state preparation, i.e., before the steering protocol starts and any input is generated. The situation is analogous to that of local filters in the Bell experiments[54]. We showed that a seemingly abstract concept, i.e., the steering-equivalent-observables measurement assemblage, or SEO, introduced to formally map a steering problem into an incompatibility one, has a direct physical interpretation. In fact, the SEOs characterize equivalence classes with respect to $\mathrm{LF}_1$ filters, and its incompatibility corresponds to the maximal steerability, quantified by the steering robustness, which can be extracted from a given assemblage via local filters. Moreover, we showed that the existence of an $\mathrm{LF}_1$ transformation in one direction implies the existence of the reverse transformation. In addition, we showed that within each equivalence class, steerability can range from (almost) zero to this maximal value.

Our results include an efficient computation of the local filter via SDP. Moreover, we showed that, given a measurement assemblage on Alice's side, there always exists a bipartite state (also efficiently computable via SDP) such that the steerability of Bob's state assemblage coincides with the incompatibility of Alice's measurement assemblage. Interestingly, the state is not necessarily maximally entangled. These results show that the previously known upper bounds for steerability, i.e., $\mathrm{SR}(\boldsymbol{\sigma}) \leq \mathrm{IR}(\boldsymbol{B}) \leq \mathrm{IR}(\boldsymbol{A})$[51], where $\boldsymbol{A}$ is the measurement assemblage on Alice's side, $\boldsymbol{\sigma}$ is the corresponding state assemblage on Bob's side, and $\boldsymbol{B}$ is the SEO, can always be saturated.

Since our filter protocol involves only local operations, we can directly compute the asymptotic conversion rate between assemblages in terms of the single-shot success probability of a single filter. We recover the theoretical results of ref. 31 and answer the open question formulated therein regarding the existence of steering dilution and the reversibility of the transformation. Finally, an explicit example of a qutrit steering distillation is also presented, which is experimentally implementable with current technology, see, e.g., refs. 55,56.

Our results naturally suggest new research directions. For instance, can we have a more general result on the quantitative relation between steerability and incompatibility within an SEO class, i.e., does the maximal steerable weight[57] in an SEO saturate the incompatible weight[58] of an SEO? Another observation is the following. Theorem 1 requires a rank constraint to guarantee the existence of the reverse transformation in LF$_1$. As we have seen in Th. 2 and Th. 3, rank constraints can be satisfied by admitting a small error, i.e., by substituting a low-rank assemblage with an arbitrarily close one of higher rank. For a given filter $K$, the construction of an approximate filter $K^\varepsilon$ admitting an inverse for a given assemblage, however, is nontrivial, as it is nontrivial in its physical and operational interpretation. We leave the question of an extension of Th. 1, including approximate transformations, to future investigation. Moreover, the conversion rate defined in Eq. (10) allows for the possibility of global operations on multiple copies of the assemblage, i.e., $\boldsymbol{\sigma}^{\otimes N}$, as is the case in entanglement theory. How can the rate be improved by using global operations? For instance, it is known that steering can be super activated when Alice performs collective measurements on many copies of the initial state[59,60] (see also the superactivation of quantum steering by two-sided local filters[61]). Therefore, our results may also be applicable beyond the resource theory of steering, e.g., when also Alice's device is partially characterized. Finally, what happens when moving from the bipartite to the multipartite scenario? It has been shown that Greenberger–Horne–Zeilinger and W-type assemblages generated by the corresponding multiparty-entangled types can be distilled by local filters[62]. Can our approach be generalized to recently proposed steering networks[63] or multiparty steering[64]? All these questions will be the object of future research.

## Methods
### Proof of Theorem 1
Proof.− First, we prove that (i) → (ii), the properties of the corresponding filter $K$ and its success probability. We denote by $\rho^{(i)}$ the reduced states for $\boldsymbol{\sigma}^{(i)}$, for $i = 1, 2$, i.e., $\rho^{(i)} = \sum_a \sigma_{a|x}^{(i)}$ and the corresponding ranges by $\mathscr{K}_{(i)} := \mathrm{ran}(\rho^{(i)})$. Using the definition of SEOs, the equivalence relation of Eq. (2), and the conventional notation of the inverse square root operator, i.e., $\rho^{-1/2} = \tilde{\rho}^{-\frac{1}{2}} \oplus 0_{\mathscr{K}^\perp}$, we can directly write

$$\sigma_{a|x}^{(1)} = \rho^{(1)1/2} U \rho^{(2)-1/2} \sigma_{a|x}^{(2)} \rho^{(2)-1/2} U^\dagger \rho^{(1)1/2}, \forall a, x. \quad (12)$$

Although the above mapping provides the correct transformation of $\boldsymbol{\sigma}^{(1)}$ to $\boldsymbol{\sigma}^{(2)}$ and is completely positive by construction, it may be nonphysical since $\rho^{(2)-1} \not\leq \mathbb{I}$, thus yielding a trace-increasing map. To obtain the correct filtering operation, it is enough to properly insert a suitable constant into the above expression. Let us first define the operator

$$\widetilde{K} := \rho^{(1)1/2} U \rho^{(2)-1/2}. \quad (13)$$

We now define the local filter in the Kraus representation in terms of a real normalization parameter $\alpha$ as

$$K := \alpha \widetilde{K} + \mathbb{I}_{\mathscr{K}_{(2)}^\perp}. \quad (14)$$

Using the condition $K^\dagger K \leq \mathbb{I}$, and denoting the maximal eigenvalue of $\widetilde{K}^\dagger \widetilde{K}$ by $\lambda_{\max}(\widetilde{K}^\dagger \widetilde{K})$, we determine the constant as

$$\alpha^2 \leq \frac{1}{\lambda_{\max}(\widetilde{K}^\dagger \widetilde{K})}. \quad (15)$$

Over all possible values, it makes sense to take $\alpha$ as real and maximal, i.e., obtaining the equality sign in Eq. (15), in order to maximize the success probability $p_{\mathrm{succ}} := \mathrm{tr}[\sum_a \sigma_{a|x}^{(2)} K^\dagger K]$ of the filtering operation. Such a probability can be directly calculated using that

$$K^\dagger K = \alpha^2 \rho^{(2)-1/2} U^\dagger \rho^{(1)} U \rho^{(2)-1/2} + \mathbb{I}_{\mathscr{K}_{(2)}^\perp}, \quad (16)$$

which, by the definition of $p_{\mathrm{succ}}$ and the cyclicity of the trace, gives

$$p_{\mathrm{succ}} = \mathrm{tr}\left[\sum_a \sigma_{a|x}^{(2)} K^\dagger K\right] = \mathrm{tr}\left[\rho^{(2)} K^\dagger K\right] = \mathrm{tr}\left[\rho^{(2)} \alpha^2 \rho^{(2)-1/2} U^\dagger \rho^{(1)} U \rho^{(2)-1/2}\right]$$
$$= \alpha^2 \mathrm{tr}\left[\rho^{(1)}\right] = \alpha^2, \quad (17)$$

which, together with Eq. (15) provides the optimal success probability.

We, then, have that

$$\sigma_{a|x}^{(1)} = \frac{K \sigma_{a|x}^{(2)} K^\dagger}{\alpha^2}. \quad (18)$$

Note that $p_{\mathrm{succ}}$ is properly normalized, since $\rho^{(2)}$ is a state and $K^\dagger K \leq \mathbb{I}$. Moreover, by construction $\rho^{(1)1/2} U \rho^{(2)-1/2}$ is zero on $\mathscr{K}_{(2)}$, so the extra identity operator does not play a role in the normalization. Also, this local filter is a valid 1W-SLOCC operation.

Finally, we notice that it is also possible to obtain an estimate of the optimal success probability directly from the eigenvalues of the reduced states $\rho^{(1)}$ and $\rho^{(2)}$. Using the facts that unitaries preserve eigenvalues [$U^\dagger \mathbb{I}_{\mathscr{K}_{(1)}} U = \mathbb{I}_{\mathscr{K}_{(2)}}$ in Eq. (2)] and that

$$\lambda_{\min}(\rho^{(1)})^{1/2} \mathbb{I}_{\mathscr{K}_{(1)}} \leq \rho^{(1)1/2} \leq \lambda_{\max}(\rho^{(1)})^{1/2} \mathbb{I}_{\mathscr{K}_{(1)}}, \quad (19)$$

$$\lambda_{\max}(\rho^{(2)})^{-1/2} \mathbb{I}_{\mathscr{K}_{(2)}} \leq \rho^{(2)-1/2} \leq \lambda_{\min}(\rho^{(2)})^{-1/2} \mathbb{I}_{\mathscr{K}_{(2)}}, \quad (20)$$

where $\lambda_{\min} > 0$ denotes the minimal nonzero eigenvalue, we can directly obtain an estimate of $\lambda_{\max}(\widetilde{K}^\dagger \widetilde{K})$ to show that

$$\frac{\lambda_{\min}(\rho^{(1)})}{\lambda_{\max}(\rho^{(2)})} \leq \lambda_{\max}(\widetilde{K}^\dagger \widetilde{K}) \leq \frac{\lambda_{\max}(\rho^{(1)})}{\lambda_{\min}(\rho^{(2)})}. \quad (21)$$

This finally gives an estimate of the success probability as

$$\frac{\lambda_{\min}(\rho^{(2)})}{\lambda_{\max}(\rho^{(1)})} \leq p_{\mathrm{succ}} \leq \frac{\lambda_{\max}(\rho^{(2)})}{\lambda_{\min}(\rho^{(1)})}. \quad (22)$$

Let us now prove that (ii) $\Rightarrow$ (iii). First, we recall the definition of the canonical representative of the equivalence class associated with the SEO **B**, namely

$$\sigma_{a|x}^{\mathbf{B}} := B_{a|x}/d, \quad (23)$$

where $d = \dim(\mathrm{ran}\rho)$. Then, by the definition of an SEO, we have $\boldsymbol{\sigma}^{(i)} \xrightarrow{\mathrm{LF}_1} \boldsymbol{\sigma}^{\mathbf{B}^{(i)}}$ and $\boldsymbol{\sigma}^{\mathbf{B}^{(i)}} \xrightarrow{\mathrm{LF}_1} \boldsymbol{\sigma}^{(i)}$ with $\mathbf{B}^{(i)}$ denoting the SEO of the assemblage $\boldsymbol{\sigma}^{(i)}$. Composing these transformation, we have the maps $\boldsymbol{\sigma}^{\mathbf{B}^{(i)}} \xrightarrow{\mathrm{LF}_1} \boldsymbol{\sigma}^{\mathbf{B}^{(j)}}$ for $(i, j) = (1, 2), (2, 1)$. Since all transformations are in LF$_1$, their composition is also in LF$_1$. For convenience, we write everything

in the global space $\mathscr{H}$, as

$$B_{a|x}^{(1)}/d_1 \oplus 0_{\mathscr{K}_{(1)}^\perp} = \frac{K(B_{a|x}^{(2)}/d_2 \oplus 0_{\mathscr{K}_{(2)}^\perp})K^\dagger}{p_{\mathrm{succ}}^{(2)}} \;\forall a,x, \quad \text{and}$$

$$B_{a|x}^{(2)}/d_2 \oplus 0_{\mathscr{K}_{(2)}^\perp} = \frac{\widetilde{K}(B_{a|x}^{(1)}/d_1 \oplus 0_{\mathscr{K}_{(1)}^\perp})\widetilde{K}^\dagger}{p_{\mathrm{succ}}^{(1)}} \;\forall a,x, \tag{24}$$

for some $K,\widetilde{K} \in \mathrm{LF}_1$ and where $p_{\mathrm{succ}}^{(1)} = \mathrm{tr}[\mathbb{I}_{\mathscr{K}_{(1)}}\widetilde{K}^\dagger\widetilde{K}]/d_1$ and $p_{\mathrm{succ}}^{(2)} = \mathrm{tr}[\mathbb{I}_{\mathscr{K}_{(2)}}K^\dagger K]/d_2$ are the corresponding success probabilities. By the conditions $\sum_a B_{a|x}^{(i)} = \mathbb{I}_{\mathscr{K}_{(i)}}$, we have

$$\mathbb{I}_{\mathscr{K}_{(1)}}/d_1 \oplus 0_{\mathscr{K}_{(1)}^\perp} = \frac{K(\mathbb{I}_{\mathscr{K}_{(2)}}/d_2 \oplus 0_{\mathscr{K}_{(2)}^\perp})K^\dagger}{p_{\mathrm{succ}}^{(2)}}, \quad \text{and}$$

$$\mathbb{I}_{\mathscr{K}_{(2)}}/d_2 \oplus 0_{\mathscr{K}_{(2)}^\perp} = \frac{\widetilde{K}(\mathbb{I}_{\mathscr{K}_{(1)}}/d_1 \oplus 0_{\mathscr{K}_{(1)}^\perp})\widetilde{K}^\dagger}{p_{\mathrm{succ}}^{(1)}}, \tag{25}$$

Using the fact that for any pair of linear maps $A$, $B$ $\dim \mathrm{ran}(AB) \le \min\{\dim \mathrm{ran}(A), \dim \mathrm{ran}(B)\}$, we obtain the two inequalities $d_2 \le d_1$ and $d_1 \le d_2$, since $d_i = \dim \mathscr{K}_i = \dim \mathrm{ran}(\mathbb{I}_{\mathscr{K}_{(i)}})$. This implies $d_1 = d_2$ and concludes this part of the proof.

Let us now prove that (iii) $\Rightarrow$ (i). By assumption, we have the transformation $\sigma^{(2)} \xrightarrow{\mathrm{LF}_1} \sigma^{(1)}$, which, combined with the definition of SEO as above, gives us the transformation $\sigma^{\mathbf{B}^{(2)}} \xrightarrow{\mathrm{LF}_1} \sigma^{\mathbf{B}^{(1)}}$. By assumption, $d_1 = d_2$, hence,

$$B_{a|x}^{(1)} \oplus 0_{\mathscr{K}_{(1)}^\perp} = \frac{K\left(B_{a|x}^{(2)} \oplus 0_{\mathscr{K}_{(2)}^\perp}\right)K^\dagger}{p_{\mathrm{succ}}^{(2)}} \;\forall a,x, \tag{26}$$

which, summing over $a$ and splitting $p_{\mathrm{succ}}^{(2)}$, gives

$$\mathbb{I}_{\mathscr{K}_{(1)}} \oplus 0_{\mathscr{K}_{(1)}^\perp} = \frac{K}{\sqrt{p_{\mathrm{succ}}^{(2)}}}\left(\mathbb{I}_{\mathscr{K}_{(2)}} \oplus 0_{\mathscr{K}_{(2)}^\perp}\right)\frac{K^\dagger}{\sqrt{p_{\mathrm{succ}}^{(2)}}}. \tag{27}$$

Let us define the map $V := K^\dagger|_{\mathscr{K}_{(1)}}/\sqrt{p_{\mathrm{succ}}^{(2)}}$, i.e., $K^\dagger$ renormalized and restricted on the subspace $\mathscr{K}_{(1)}$. We have that $V : \mathscr{K}_1 \to \mathscr{K}_2$ is an isometry, since $V^\dagger V = \mathbb{I}_{\mathscr{K}_{(1)}}$. As an isometry, $V$ is injective, and since $d_1 = d_2$ it is also surjective. This implies that $V$ is a unitary between $\mathscr{K}_{(1)}$ and $\mathscr{K}_{(2)}$. Similarly, one obtains that $V^\dagger$ is a unitary from $\mathscr{K}_{(2)}$ to $\mathscr{K}_{(1)}$. Hence, $V$ can then be extended to a global unitary $U : \mathscr{H} \to \mathscr{H}$, simply by completing it with a mapping from an orthonormal bases of $\mathscr{K}_{(2)}^\perp$ to an orthonormal basis $\mathscr{K}_{(1)}^\perp$. We then have

$$B_{a|x}^{(1)} \oplus 0_{\mathscr{K}_{(1)}^\perp} = U\left(B_{a|x}^{(2)} \oplus 0_{\mathscr{K}_{(2)}^\perp}\right)U^\dagger \quad \forall a,x, \tag{28}$$

which concludes the proof of the implication (iii) $\Rightarrow$ (i).

To conclude the proof, the only thing left to prove is that the transformation is provably optimal if there are sufficient linearly independent elements in the original state assemblage to completely characterize the channel. The idea is relatively simple and is based on the fact that, under this condition, the transformation is uniquely defined. By contradiction, let us assume we have another optimal transformation $K_0$ over $\mathrm{LF}_1$, mapping $\sigma^{(2)} \to \sigma^{(1)}$ and that $\{\sigma_{a|x}^{(2)}\}$ consists of at least $d^2$ linearly independent elements. We have

$$\sigma_{a|x}^{(1)} = \frac{K_0 \sigma_{a|x}^{(2)} K_0^\dagger}{p_{\mathrm{succ}}} = \rho^{(1)1/2} U \rho^{(2)-1/2} \sigma_{a|x}^{(2)} \rho^{(2)-1/2} U^\dagger \rho^{(1)1/2}, \forall a,x. \tag{29}$$

Since sufficient linearly independent subnormalized states $\sigma_{a|x}^{(2)}$ are available in order to characterize the filter, this implies that $K_0/\sqrt{p_{\mathrm{succ}}} = \rho^{(1)1/2} U \rho^{(2)-1/2}$. In fact, note that $K_0(\cdot)K^\dagger : \mathcal{L}(\mathscr{H}) \to \mathcal{L}(\mathscr{H})$ is

a linear map from linear operators to linear operators. Thus, it is completely characterized by its action on a basis, i.e., $d^2$ linearly independent linear operators, where $d$ is the dimension of $\mathscr{H}$ and, thus, $d^2$ is the dimension of $\mathcal{L}(\mathscr{H})$. Finally, since $K_0$ is the filtering maximizing the success probability, we have $K_0 K_0^\dagger \le \mathbb{I}$ and $K_0 K_0^\dagger \nleq (1-\varepsilon)\mathbb{I}$ for all $\varepsilon > 0$. This corresponds to the choice of the maximum $\alpha$ in Eq. (15). This concludes the proof.

As a final note, since $\sim_{\mathrm{SEO}}$ and $\sim_{\mathrm{LF}_1}$ are symmetric, the roles of $\sigma^{(1)}$ and $\sigma^{(2)}$ can be exchanged in Theorem 1 (iii).

## Proof of Theorem 2

The first observation is that, up to an embedding and a change of local basis (i.e., adding a $\oplus 0_{\mathscr{K}^\perp}$ and a unitary $U$, as in Eq. (2)), a generic element $\sigma$ in the equivalence class of the SEO $\mathbf{B}$ can be obtained by the representative $\sigma^{\mathbf{B}}$ as

$$\sigma_{a|x} = \eta^{1/2} B_{a|x} \eta^{1/2} = d\eta^{1/2} \sigma_{a|x}^{\mathbf{B}} \eta^{1/2}, \tag{30}$$

for some full-rank reduced state $\eta$. We can now proceed to prove Theorem 2.

Proof of the supremum—First, it is useful to recall the dual SDP formulations of IR and SR (see respectively refs. [46],[65]):

$$\begin{aligned} \text{Given} \quad & \mathbf{B} \\ \text{Find} \quad & \max_{\boldsymbol{\omega},\eta} \mathrm{tr}\left(\sum_{a,x} \omega_{a|x} B_{a|x}\right) =: 1 + \mathrm{IR}(\mathbf{B}) \\ \text{s.t.} \quad & \eta \ge \sum_{a,x} D(a|x,\lambda)\omega_{a|x}, \forall \lambda, \\ & \omega_{a|x} \ge 0, \; \mathrm{tr}(\eta) = 1, \end{aligned} \tag{31}$$

and

$$\begin{aligned} \text{Given} \quad & \boldsymbol{\sigma} \\ \text{Find} \quad & \max_{\mathbf{F}} \mathrm{tr}\left(\sum_{a,x} F_{a|x} \sigma_{a|x}\right) =: 1 + \mathrm{SR}(\boldsymbol{\sigma}) \\ \text{s.t.} \quad & \mathbb{I} \ge \sum_{a,x} D(a|x,\lambda)F_{a|x}, \forall \lambda, \\ & F_{a|x} \ge 0. \end{aligned} \tag{32}$$

Here, $D(a|x, \lambda)$ is the deterministic postprocessing of $a$ with respect to $x$, $\lambda$ appearing in the primal problem, i.e., $\delta_{a,\lambda_x}$. Notice that we can interpret $\eta$ as a valid quantum state and $\boldsymbol{\omega}$ and $\mathbf{F}$ as the incompatibility witnesses and steering witnesses, respectively.

By Theorem 1, we can associate the SEO $\mathbf{B}$ to the equivalence class $[\boldsymbol{\sigma}]$, w.r.t. $\sim_{\mathrm{LF}_1}$. This implies that $\boldsymbol{\sigma} \sim_{\mathrm{SEO}} \boldsymbol{\sigma}^{\mathbf{B}}$ and $\boldsymbol{\sigma} \sim_{\mathrm{LF}_1} \boldsymbol{\sigma}^{\mathbf{B}}$, where $\boldsymbol{\sigma}^{\mathbf{B}}$ is defined in Eq. (23). We also recall that a generic element of the equivalence class can be written as $\sigma_{a|x} = \eta^{1/2} B_{a|x} \eta^{1/2}$ for some full-rank state $\eta$ (see Eq. (30)).

By combining the definition of $\mathrm{SR}^{\mathrm{sup}}$ with Theorem 1 and Eq. (32), we can upper bound the maximal steering robustness over all SEO-equivalent assemblages via the following optimization problem

$$\begin{aligned} \text{Given} \quad & \mathbf{B} \\ \text{Find} \quad & \max_{\mathbf{F},\eta} \mathrm{tr}\left(\sum_{a,x} F_{a|x} \eta^{1/2} B_{a|x} \eta^{1/2}\right) =: 1 + \Omega \\ \text{s.t.} \quad & \mathbb{I} \ge \sum_{a,x} D(a|x,\lambda)F_{a|x}, \forall \lambda, \\ & F_{a|x} \ge 0, \quad \mathrm{tr}(\eta) = 1. \end{aligned} \tag{33}$$

Notice that the problem in Eq. (33) is no longer an SDP, since it contains as an objective function that is nonlinear in $\eta$ and $\mathbf{F}$. Nevertheless, we can now show that every feasible solution of the SDP in Eq. (31) is a feasible solution of the problem in Eq. (33) and vice versa. In fact, given $\boldsymbol{\omega}$, $\eta$ feasible solution of Eq. (31), we can define $F_{a|x} := \eta^{-1/2} \omega_{a|x} \eta^{-1/2}$, which satisfies $F_{a|x} \ge 0$ and $\mathbb{I} \ge \sum_{a,x} D(a|x,\lambda)F_{a|x}$, even when $\eta$ is not full-

rank and we invert it on just a subspace. Conversely, given $F$, $\eta$ feasible solution of Eq. (33), we can define $\omega_{a|x} := \eta^{1/2} F_{a|x} \eta^{1/2}$, which satisfies $\omega_{a|x} \geq 0$ and $\eta \geq \sum_{a,x} D(a|x,\lambda) \omega_{a|x}$. Again, no problem arises if $\eta$ is not full-rank. Finally, it is clear that this construction provides the same value for the objective function in both directions. We have thus proven that each solution to one problem provides a solution to the other, without changing the objective function, which implies that the optimal value is the same.

Finally, we need to verify that the optimal solution $\Omega$ is indeed the supremum over all assemblages in the same equivalence class, i.e., $\Omega = \mathrm{SR}^{\sup}([\boldsymbol{\sigma}])$. The missing condition comes from the fact that if the state $\eta$ is not full-rank, then the constructed assemblage is not in the same class as $\boldsymbol{\sigma}^{\mathbf{B}}$. However, for any state $\eta$, we can always find another state $\tilde{\eta}$ that is arbitrarily close to it. Let us define $\Pi_\eta$ as the projector on the range of $\eta$ with rank $r := \mathrm{tr}[\Pi_\eta] < d$. For any $\varepsilon > 0$, there exists $\delta$ such that we can approximate the solution of the problem in Eq. (33) up to $\varepsilon$ via the following construction. First, we construct a full-rank $\eta_\varepsilon$ as

$$\eta_\varepsilon := (1-\delta)\eta + \frac{\delta}{(d-r)}(\mathbb{I} - \Pi_\eta), \tag{34}$$

which approximates the optimal value $\Omega$ in Eq. (33) as $|\Omega(\boldsymbol{F}, \eta) - \Omega(\boldsymbol{F}, \eta_\varepsilon)| \leq \varepsilon$. Similarly, we define $\omega'_{a|x} = (1-\delta)\omega_{a|x}$ to preserve the condition in Eq. (31). This guarantees that we still obtain a feasible solution.

This solution approximates the optimal value that follows directly from the continuity of the objective function in Eq. (33). A concrete estimate for $\delta$ can be obtained by estimating the Hilbert–Schmidt norm of the difference

$$\eta^{1/2} B_{a|x} \eta^{1/2} - \eta_\varepsilon^{1/2} B_{a|x} \eta_\varepsilon^{1/2}$$
$$= \delta \left[ \eta^{1/2} B_{a|x} \eta^{1/2} - \frac{1}{d-r} \left( (\mathbb{I} - \Pi_\eta) B_{a|x} \eta^{1/2} + \eta^{1/2} B_{a|x} (\mathbb{I} - \Pi_\eta) \right) \right] + O(\delta^2), \tag{35}$$

and applying the Cauchy–Schwarz inequality to the objective function, using also the fact that $0 \leq B_{a|x}, F_{a|x} \leq \mathbb{I}$ to upper bound their norm. It is then clear that the SDP in Eq. (33) provides a vanishing upper bound in the limit $\delta \to 0$ for the difference between the optimal value in the problem and that obtained by the substitution $\eta \to \eta_\varepsilon$. This shows that IR is indeed the supremum and concludes this part of the proof.

*Proof of the infimum—* To prove the infimum, we consider that for every measurement assemblage $\boldsymbol{A} = \boldsymbol{B}^T$ and any full-Schmidt-rank state $|\psi\rangle = \sum_{i=1}^{d} \mu_i |ii\rangle$, there exists an assemblage $\boldsymbol{\sigma}$ such that[43]

$$\sigma_{a|x} = \mathrm{tr}_A[|\psi\rangle\langle\psi| B_{a|x}^T \otimes \mathbb{I}] = \tau^{1/2} B_{a|x} \tau^{1/2}, \tag{36}$$

where $T$ denotes the transpose in the basis $\{|i\rangle\}$ appearing in the Schmidt decomposition of $|\psi\rangle$, and $\tau = \sum_{i=1}^{d} \mu_i^2 |i\rangle\langle i|$ is the reduced state of $|\psi\rangle$. In other words, for any full-Schmidt-rank state $|\psi\rangle$ and any measurement assemblage $\boldsymbol{B}^T$, we can obtain a state assemblage $\boldsymbol{\sigma}$ that gives $\boldsymbol{B}$ as its SEO. In particular, this implies that for any assemblage $\boldsymbol{\sigma}$, we can find $\boldsymbol{\sigma}'$ such that $\boldsymbol{\sigma} \sim_{\mathrm{SEO}} \boldsymbol{\sigma}'$ and $\boldsymbol{\sigma}'$ comes from a quantum state $|\psi\rangle$ with arbitrarily low entanglement.

Now, we consider another fact about steering robustness[46], namely, that $\mathrm{ER}_g(|\psi\rangle) \geq \mathrm{SR}(\boldsymbol{\sigma})$ with $\mathrm{ER}_g(|\psi\rangle)$ being the generalized entanglement robustness of $|\psi\rangle$ (see ref. 66 for more details). In turn, $\mathrm{ER}_g(|\psi\rangle)$ is upper bounded by the random entanglement robustness $\mathrm{ER}_r(|\psi\rangle)$, obtained when mixing with the maximally mixed state. For pure states, this has a simple expression in terms of the Schmidt decomposition $|\psi\rangle = \sum_{i}^{d} \mu_i |ii\rangle$, where the vectors are ordered such that $\mu_1 \geq \mu_2 \geq \mu_3 \ldots \geq 0$, namely[67]:

$$\mathrm{ER}_r(|\psi\rangle) = \mu_1 \mu_2 \mathrm{d}_A \mathrm{d}_B. \tag{37}$$

For any $\varepsilon > 0$, we can take $|\psi\rangle = \sqrt{1 - (d-1)\varepsilon'}|00\rangle + \sqrt{\varepsilon'}|11\rangle + \ldots \sqrt{\varepsilon'}|d-1,d-1\rangle$, with $\varepsilon' < \varepsilon^2/(d_A d_B)^2$. This gives $\mu_1 = \sqrt{1 - \varepsilon'} < 1$ and $\mu_2 = \sqrt{\varepsilon'} < \varepsilon/(d_A d_B)$; hence, $\mathrm{ER}_r(|\psi\rangle) \leq \varepsilon$. Since $\varepsilon$ is arbitrary, the infimum is zero, which concludes the proof.

## Data availability

Dataset sharing is not applicable to this article as no data sets were generated or analyzed during this study.

## Code availability

Source codes of the plots are available from the authors upon request.

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

## Acknowledgements

The authors acknowledge fruitful discussions with Yi-Te Huang, Yeong-Cherng Liang, and Gelo Noel M Tabia. This work is supported partially by the National Science and Technology Council, Taiwan (Grants Nos. MOST 110-2811-M-006-546, MOST 111-2917-I-564-005, MOST 111-2112-M-005-007-MY4, and MOST 111-2123-M-006-001) and the Army Research

Office (under Grant No. W911NF-19-1-0081). C.-Y.H. is supported by ICFOstepstone (the Marie Skłodowska-Curie Co-fund GA665884), the Spanish MINECO (Severo Ochoa SEV-2015-0522), the Government of Spain (FIS2020-TRANQI and Severo Ochoa CEX2019-000910-S), Fundació Cellex, Fundació Mir-Puig, Generalitat de Catalunya (SGR1381 and CERCA Program), the ERC AdG CERQUTE, and the AXA Chair in Quantum Information Science. C.B. is supported by the Austrian Science Fund through Projects No. ZK 3 (Zukunftskolleg) and No. F7113 (BeyondC).

## Author contributions

H.-Y.K., C.-Y.H., S.-L.C., and C.B. conceived the research and proved the theoretical results. Y.-N.C. and C.B. supervised the research and were responsible for the integration among different research units. All authors contributed to the discussion of the central ideas and to the writing of the manuscript.

## Competing interests

The authors declare no competing interests.
