## [Peer Review File · Nature Communications]

Complete classification of steerability under local filters and its relation with measurement incompatibility.REVIEWER COMMENTS

Reviewer #1 (Remarks to the Author):

In this article, the authors study the scenario of bipartite steering + local filtering on the trusted party. This is equivalent to the 1-way LOCC class of operations, relevant for one-sided device-independent protocols (bipartite protocols, eg QKD, where the device of Bob is trusted but not the one of Alice).

The first key result is an iff relation between SEO equivalence and filtering equivalence (that is, two assemblages have the same SEO up to a unitary iff they can be transformed into each other with a local filter), allowing for an explicit computation of such filters.

The second result consists of upper and lower bounds of the steering robustness of assemblages in the filtering class of equivalence: the lower bound is 0 and the upper bound is the incompatibility robustness of the SEO. Moreover, these bounds are tight and can be reached via an SDP computation of the local filter.

Finally, the article provide a proof that for any set of POVMs there exists a bipartite state such that the resulting assemblage has steering robustness equal to the incompatibility robustness of the POVMs. Moreover, that state can be found via an SDP.

I believe the results are correct, modulo a technicality I'm unsure about, see question 1 below.

As far as I'm aware of, these results are all novel and non-trivial. They allow linking nicely and elegantly two different concepts that don't seem directly in such a close relation.

As often, linking two concepts is fruitful not only from a fundamental standpoint, but also from a practical one: it lets us importing known tools from one side of the problem to the linked side. Here, for instance, it allows us to easily check whether an assemblage can be transformed into another one via local filtering, as well as computing the local filter in question.

As one-sided device-independent protocols are within application reach, I expect the results presented here to be relevant and useful.

I do have a few questions/comments to the authors:

1- You claim that local filtering transformations are always reversible, but I fail to see how it's the case when the said filter is not full rank. To take the extremal case: starting with the singlet $|00\rangle + |11\rangle$ I can filter it to $|00\rangle$, but cannot filter it back to the singlet. I'm not sure that I understood how you prevented these cases in your theorem/proof, as you seem to allow any local filters, even non-full-rank ones.

2- Isn't the IR of the SEO equal to the SR of the assemblage defined by SEO/d ? If so, doesn't your 'Theorem 2' imply that the best filter is the one bringing the assemblage to SEO/d ?

3- You've studied a particular measure of steering and incompatibility, namely the (generalized) robustness. What happens to theorem 2 and 3 for different measures of steering/incompatibility? Is there an already known measure that would be constant within an SEO equivalence class?

4- My intuition for Theorem 1 is the following: the SEO defines a kind of canonical form for an assemblage, where the bipartite state used has maximally mixed marginal on Bob's side, which is exactly the freedom you have using a local filter on Bob. Hence, same SEO class means same filtering class (same freedom from canonical form). Does that framing make sense? If so, is it worth including it in the article (sincere question)?

More generally, I'm not sure to fully grasp the link between an assemblage and its SEO (besides the iff link between incompatibility of SEO and steerability of assemblage). I reckon some expansion on that aspect would be valuable.

Reviewer #2 (Remarks to the Author):

In the submitted paper the authors investigate the possibility of transforming of one steering assemblage into another via post-selection. The article is timely and interesting since steering is currently an active research area, both theoretically and experimentally. It is shown that the equivalence classes due to post-selection on Bob's side coincide with the equivalence classes given by the steering equivalent observables (SEO). This is a nice result and could be of potential use. The authors also show that the equivalence class contains assemblages with steering robustness varying from 0 to the upper bound given by SEO and Alice's observables. This is to be expected, as it is well known that post-selection can increase or decrease violations of Bell inequalities and so one can expect the same for steering. It is questionable whether Bob would actually want to perform post-selection as this can create loophole, just like in the case of Bell nonlocality. Given all of these fact I believe that the results should be published, but I am inclined to think that the results may be more suitable to a specialized physics journal.

The paper is relatively clearly written, but several parts of it can be improved:

- The definition of steering robustness is correct, but the authors should explain why this coincides with the standard definition that $1/(1+t)$ ($\sigma_{a|x} + t \tau_{a|x}$) has local hidden state model.
- The authors should address the case when the success probability in (3) is zero.
- Theorem 1. should be split into 2 statements, one for each implication. In the current form it is hard to understand because most of the text discusses only one of the implications, since the other is trivial. Also λ_{\max} is the maximum eigenvalue of what operator?
- In Theorem 3. it is not clear whether the assemblage is generated by the given measurements.

Resubmission of the manuscript “Complete classification of steerability under local filters and its relation with measurement incompatibility” NCOMMS-22-03113

REPLY TO THE REVIEWERS

Reply to Reviewer #1

First of all, we would like to thank again Reviewer #1 for their comments on the manuscript. They helped us to rethink the significance of our results, expand them, and improve their presentation. Below we provide a detailed answer to all the points raised.

The reviewer writes:

In this article, the authors study the scenario of bipartite steering + local filtering on the trusted party. This is equivalent to the 1-way LOCC class of operations, relevant for one-sided device-independent protocols (bipartite protocols, eg QKD, where the device of Bob is trusted but not the one of Alice).

The first key result is an iff relation between SEO equivalence and filtering equivalence (that is, two assemblages have the same SEO up to a unitary iff they can be transformed into each other with a local filter), allowing for an explicit computation of such filters.

The second result consists of upper and lower bounds of the steering robustness of assemblages in the filtering class of equivalence: the lower bound is 0 and the upper bound is the incompatibility robustness of the SEO. Moreover, these bounds are tight and can be reached via an SDP computation of the local filter.

Finally, the article provide a proof that for any set of POVMs there exists a bipartite state such that the resulting assemblage has steering robustness equal to the incompatibility robustness of the POVMs. Moreover, that state can be found via an SDP.

I believe the results are correct, modulo a technicality I’m unsure about, see question 1 below.

Our reply:

We thank the referee for the accurate summary of our paper and the points raised.

The reviewer writes:

As far as I’m aware of, these results are all novel and non-trivial. They allow linking nicely and elegantly two different concepts that don’t seem directly in such a close relation.

As often, linking two concepts is fruitful not only from a fundamental standpoint, but also from a practical one: it lets us importing known tools from one side of the problem to the linked side. Here, for instance, it allows us to easily check whether an assemblage can be transformed into another one via local filtering, as well as computing the local filter in question.

As one-sided device-independent protocols are within application reach, I expect the results presented here to be relevant and useful.

Our reply:

We thank the reviewer for appreciating the value and significance of our results and their positive comments on our manuscript.

The reviewer writes:

I do have a few questions/comments to the authors:

1- You claim that local filtering transformations are always reversible, but I fail to see how it’s the case when the said filter is not full rank. To take the extremal case: starting with the singlet $|00\rangle + |11\rangle$ I can filter it to $|00\rangle$, but cannot filter it back to the singlet. I’m not sure that I understood how you prevented these cases in your theorem/proof, as you seem to allow any local filters, even non-full-rank ones.

Our reply:

We thank the reviewer for this invaluable comment. Indeed the reviewer is correct. This makes us rethink the proof of Th. 1, where we found an implicit assumption. We are then able to correct the statement. For completeness, we should say that our result concerns the transformation of assemblages, rather than the transformation of bipartite states. However, an example concerning the corresponding assemblages would indeed work. The implicit assumption in our previous (erroneous) proof of Th. 1 concerns the rank of the two reduced states, $\rho^{(1)}$ and $\rho^{(2)}$ associated with the assemblages $\vec{\sigma}^{(1)}$, and $\vec{\sigma}^{(2)}$. In the case of a transformation only in one direction, we can (now) claim that the reverse transformation exists only if this transformation preserves the rank of the reduced state (see statement (iii) of Th. 1 in the revised manuscript). For completeness, we should also say that local filters do not need to be full rank. It is necessary only that a local filter preserves the range of the reduced state for Bob. Finally, we admit that speaking

of “reversible transformations” for non-full-rank filters is improper. The transformation *per se* is not reversible; what can be reversed is the mapping from one assemblage to the other, provided that the rank condition above is satisfied.

To correct this, we revised the manuscript, the statement of Th. 1, its proof, and the discussion below it.

Finally, we would like to comment on the fact that, as discussed extensively in Th. 2 and 3 of our manuscript, the rank constraint can often be circumvented by admitting a small amount of noise ε . In fact, for any low-rank assemblage, there exists a higher rank one that is arbitrarily close (in any norm). Of course, this still does not mean that we can invert the original filter (e.g., from $|00\rangle + |11\rangle$ to $|00\rangle$), but that the original filter can be approximated by an invertible one. We obtain some partial results in this direction, but several questions are left open. For instance, several constructions are possible, e.g., starting from the approximation of the filter or starting from the approximation of the target assemblage. At the present moment, it is not clear what are the meaningful constraints to impose for this approximate realization and what are their physical and operational interpretations. For all these reasons, we believe we should take some time to further explore this problem and present it in a separate publication. This future research direction is briefly presented in the “Discussion” section.

The reviewer writes:

2- Isn't the IR of the SEO equal to the SR of the assemblage defined by SEO/d ? If so, doesn't your 'Theorem 2' imply that the best filter is the one bringing the assemblage to SEO/d ?

Our reply:

We thank the reviewer for this comment. This is a very interesting point that were not emphasized enough. Indeed, one may be tempted to formulate this hypothesis, given the many results in the literature that use the maximally entangled state to show the maximal steerability Refs. [46, 52, and 57] and the applications of the one-sided device-independent protocols in Refs. [16, 17, 54, and 55]. In addition, Nery *et al.*, in Ref. [31] show that the singlet assemblage (the maximally entangled state + mutually unbiased bases) is the most steerable in their scenario.

Our results show that the IR of the SEO is *not* the SR of the assemblage defined by SEO/d . We provide an explicit example of this, found by numerical search, in Sect. 3 of the Supplemental Information. Even though it was not the goal of the example, one can easily verify that $\text{IR}(A) > \text{SR}(A/d)$. The intuition can be easily understood as follows. IR and SR have a similar definition; however, the noise term in IR must be measurement assemblages, in which each POVM sums up to the identity, whereas the noise term in SR must be a state assemblages, so each ensemble must sum up to a reduced state. The latter is, consequently, a less constrained problem, which gives $\text{IR} \geq \text{SR}$. What can happen is that the original assemblage in SR and the noise term do not sum up to the same reduced state, which forbids the use of Eq. (1) to directly transform the SR problem into the IR problem.

One of the surprising aspects of our results is, indeed, the fact that despite these differences among the two problems, local filters are still able to provide assemblages saturating the upper bound $\text{IR}(\vec{B}) \geq \text{SR}(\vec{\sigma})$. To further emphasize the fact that this intuition is misleading, we add a discussion below Theorem 2.

The reviewer writes:

3- You've studied a particular measure of steering and incompatibility, namely the (generalized) robustness. What happens to theorem 2 and 3 for different measures of steering/incompatibility? Is there an already known measure that would be constant within an SEO equivalence class?

Our reply:

This is indeed a very good question. We believe that the steerability and incompatibility are quantitative equivalent when considering the robustness-based measures, i.e., reduced-state robustness and LHS steering robustness in Ref. [52]. It is because we just need to insert additional constraints into the Eqs. (31) and (32). We also have some partial results for the weight-based measures, i.e., steerable weight proposed in Skrzypczyk *et al.* [PRL **112**, 180404 (2014), Ref. 57]. The main difficulty is that the constraint in the dual SDP formulations of incompatible weight does not guarantee $\eta > 0$ (cf., Eq. (31)). Therefore, the feasible solution of η is not a valid quantum state such that it cannot be inserted into Eq. (30). In addition, a more general approach from the perspective of resource theory is also ambiguous. It is known that some incompatibility measures provide upper bounds on the associated steering measures, i.e., $\text{IR} \geq \text{SR}$ and $\text{IW} \geq \text{SW}$. However, the general relation is still unknown. Both problems are nontrivial, and we still have not gotten conclusive results in these directions. For these reasons, we decide not to include such results in the manuscript, but these problems will definitely be the objectives of future research. We add a comment in the section “Discussion”.

Regarding to the last comment, we admit that, to our understanding, there is no steering measure which is a constant within an SEO equivalence class. The only particular case is that for given two maximal projective von Neumann measurements, all pure entangled states provide the maximal steerable weight ($\text{SW}=1$). This result is presented in PRL **112**, 180404 (2014) [Ref. 57]. However, for all other cases this remains an open question.

The reviewer writes:

4- My intuition for Theorem 1 is the following: the SEO defines a kind of canonical form for an assemblage, where the bipartite state used has maximally mixed marginal on Bob's side, which is exactly the freedom you have using a local filter on Bob. Hence, same SEO class means same filtering class (same freedom from canonical form). Does that framing make sense? If so, is it worth including it in the article (sincere question)? More generally, I'm not sure to fully grasp the link between an assemblage and its SEO (besides the iff link between incompatibility of SEO and steerability of assemblage). I reckon some expansion on that aspect would be valuable.

Our reply:

This framing indeed makes sense. The SEO/d assemblage can be seen as a canonical form for the assemblages in a given equivalence class. We, thus, decide to include a discussion on this aspect after Th. 1.

Reply to Reviewer #2

We would like to thank again Reviewer #2 for their comments on the manuscript. In particular, the comment about possible loopholes in the steering protocol allowed us to rethink the role of the local filters and, more in general, our results from an operational perspective. Below we provide a detailed answer to all the points raised.

The reviewer writes:

In the submitted paper the authors investigate the possibility of transforming of one steering assemblage into another via post-selection. The article is timely and interesting since steering is currently an active research area, both theoretically and experimentally. It is shown that the equivalence classes due to post-selection on Bob's side coincide with the equivalence classes given by the steering equivalent observables (SEO). This is a nice result and could be of potential use. The authors also show that the equivalence class contains assemblages with steering robustness varying from 0 to the upper bound given by SEO and Alice's observables.

Our reply:

We thank the reviewer for the accurate summary and the positive comments on our results.

The reviewer writes:

This is to be expected, as it is well known that post-selection can increase or decrease violations of Bell inequalities and so one can expect the same for steering.

Our reply:

We thank the reviewer for this comment as it gives us the chance to further emphasize the significance of our results. It may be true that one could expect steerability to increase or decrease as a consequence of local filtering. Our work, however, goes much beyond this qualitative statement. We provide exact upper and lower bounds on the amount of steerability that can be obtained via local filters. Moreover, we do not only provide this value, but the exact protocol, i.e., the set of local filters, to achieve that, together with their success probabilities. Finally, all these quantities are efficiently computable via SDPs.

In our opinion, the correct comparison is not with the postselection of experimental outcomes, but rather a post-selection at the level of the bipartite state preparation. In fact, all the filtering procedures are concluded before the random input x for Alice is generated. Moreover, these types of transformations arise in the context of the resource theory of steering and should be interpreted there. Similarly, transformations of entanglement (i.e., entanglement distillation and dilution) are interpreted in terms of the resource theory of entanglement (LOCC operations) and transformation in Bell nonlocality are interpreted in terms of *local wirings*. To our understanding, current results on Bell nonlocality are relatively limited and do not provide a general solution of the problem comparable to the one presented in the present manuscript for steering. For instance, there is no general way to find out the optimal wiring which transforms any correlation to the one violating the Bell inequality maximally or violating the Bell inequality maximally within a class (see, e.g., PRA 73, 012101 (2006), PRL 102, 120401 (2009), and PRA 100, 012102 (2019)).

Finally, we would like to remark that the problem of equivalence classes and transformation among resources was formulated in the original paper by Gallego and Aolita about the resource theory of steering, i.e., Ref. [19]. In that paper, only a special result for the case of pure-state assemblages was derived. Further work by Nery *et al.* in Ref. [31] analyzed the case of assemblages arising from a two-qubit pure state. We provide a complete solution to the problem. Moreover, we address two open questions given by Nery *et al.* at the end of their paper, namely, the existence of a dilution process and the conditions for the reversibility of the transformation, now formulated simply in terms of the rank of the reduced state (see the new version of Th. 1).

The reviewer writes:

It is questionable whether Bob would actually want to perform post-selection as this can create loophole, just like in the case of Bell nonlocality. Given all of these fact I believe that the results should be published, but I am inclined to think that the results may be more suitable to a specialized physics journal.

Our reply:

We thank the reviewer for this comment that allows us to further elaborate and clarify the relationship between local filters and loopholes in steering experiments. The loopholes associated with a steering experiment are, essentially, those also associated with Bell experiments. For instance, one can look at the 2012 Vienna experiment [Wittmann et al. New J. Phys. 14 053030 2012] or the 2020 NIST experiment [Mazurek et al. Conference on Lasers and Electro-Optics, OSA]. These can be briefly summarized as follows:

- locality loophole: failure to provide a space-like separation between the input generation event on one side and the output generation event on the other.
- (detector) efficiency loophole: failure to produce (“often enough”) an output for a given input (see also [Bennet et al. PRX 2, 031003 (2012)] and [Slussarenko et al. npj Quantum Inf. 8, 20 (2022)]).

- freedom-of-choice loophole: failure to provide independent sources of randomness for the generation of the inputs.

As it is clear from the above summary, all these loopholes concern what happens *after* the random inputs have been generated. Our filtering operation, however, can be performed well before any input is generated and the steering protocol is initiated. In this sense, the local filter is more reminiscent of a state preparation, such as the heralded entanglement generation, based on a postselection of entangled states. The heralded entanglement generation is commonly used in, e.g., the loophole-free Bell experiments in Delft [Hensen et al. *Nature* 526, 682-686 (2015)] and Munich [Rosenfeld et al. *PRL* 119, 010402 (2017)].

Finally, we note that a similar argument on why a local filter does not create any loophole is presented in the introduction of Hirsch et al. in *PRL* 111, 160402 (2013). They clearly state that the local filtering operations do not introduce the detection loophole in the Bell experiment, precisely for the same reasons discussed above.

The reviewer writes:

The paper is relatively clearly written, but several parts of it can be improved: - The definition of steering robustness is correct, but the authors should explain why this coincides with the standard definition that $1/(1+t)(\sigma_{a|x} + t\tau_{a|x})$ has local hidden state model.

Our reply:

We thank the reviewer for pointing out the lack of clarity in our definition of steering robustness. As the reviewer points out, the two definitions are indeed equivalent. To simplify the discussion, we simply adopted the standard formulation.

The reviewer writes:

- The authors should address the case when the success probability in (3) is zero.

Our reply:

We thank the referee for this comment. We have included this case in the text.

The reviewer writes:

- Theorem 1. should be split into 2 statements, one for each implication. In the current form it is hard to understand because most of the text discusses only one of the implications, since the other is trivial. Also λ_{\max} is the maximum eigenvalue of what operator?

Our reply:

We thank the reviewer for this comment. We have reformulated Th. 1, also following the remarks of Reviewer #1, and separated the results into three statements. In our notation, $\lambda_{\max}(X)$ denotes the maximal eigenvalue of the operator X . We have clarified this in the text.

The reviewer writes:

- In Theorem 3. it is not clear whether the assemblage is generated by the given measurements.

Our reply:

We have emphasized that the optimal assemblage can be constructed by the given measurements and the optimal bipartite state in Theorem 3.

SUMMARY OF THE CHANGES

All the changes in the manuscript correspond to the points discussed above and are highlighted in blue. No changes have been made to the supplemental information.

REVIEWERS' COMMENTS

Reviewer #1 (Remarks to the Author):

After reading the response of the authors and the new version of the manuscript, I believe all my comments and questions have been successfully tackled.

That is, I reckon the paper is correct, clearly written, and worth being published.

Reviewer #2 (Remarks to the Author):

The authors addressed all of the comments by both reviewers and significantly improved their paper. I am especially happy about the clarification that local filter does not open a loophole, here the assessment by the authors is entirely correct and the filtering can be seen as a part of the preparation of the state. This also invalidates my previous comment and shows that the results presented in the paper are highly relevant, both theoretically and for experimental applications. Hence I recommend publication of the submitted paper.

A minor comment: the fifth sentence in Discussion should probably be "The situation is analogous to that of local *wirings* in the Bell experiments."

Resubmission of the manuscript “Complete classification of steerability under local filters and its relation with measurement incompatibility” NCOMMS-22-03113

REPLY TO THE REVIEWERS

Reply to Reviewer #1

After reading the response of the authors and the new version of the manuscript, I believe all my comments and questions have been successfully tackled.

That is, I reckon the paper is correct, clearly written, and worth being published.

Our reply:

We would like to thank the Referee for this positive report about our work. We highly appreciate this recommendation.

Reply to Reviewer #2

The authors addressed all of the comments by both reviewers and significantly improved their paper. I am especially happy about the clarification that local filter does not open a loophole, here the assessment by the authors is entirely correct and the filtering can be seen as a part of the preparation of the state. This also invalidates my previous comment and shows that the results presented in the paper are highly relevant, both theoretically and for experimental applications. Hence I recommend publication of the submitted paper.

A minor comment: the fifth sentence in Discussion should probably be "The situation is analogous to that of local *wirings* in the Bell experiments."

Our reply:

We would like to thank the Referee for their carefully reading our work manuscript and their positive opinions, in particular that "the results presented in the paper are highly relevant, both theoretically and for experimental applications". We also appreciate that they recommended it for publication.

Regarding the minor comment: Ref. 54 discusses how the local filtering operations do not open a loophole in the Bell scenario. This argument can be easily adapted to our scenario. This is the analogy we are referring to in our discussion. As such, the current formulation in our manuscript is correct. The referee may have been misled in thinking that by the discussion of the resource theory of nonlocality, which includes local wiring. This is not, however, what we are referring to in that passage.